# Contrast-Controllable Image Enhancement Based on Limited Histogram

**Xin Fan** [1,2] [iD]**, Junyan Wang** [1]**, Haifeng Wang** [2] **and Changgao Xia** [1,*]

1 School of Automotive & Traffic Engineering, Jiangsu University, Zhenjian 212013, China
2 School of Automotive & Traffic Engineering, Jiangsu University of Technology, Changzhou 213001, China
* Correspondence: xiacg@ujs.edu.cn

**Abstract:** To address the technical shortcomings of conventional histogram equalization (HE), such as over-enhancement and artifacts, we propose a histogram-constrained and contrast-tunable HE technique for digital image enhancement. Firstly, the input image histogram is partitioned into two parts, the main histogram and the constrained histogram, by a cumulative probability density threshold; second, the main histogram is redistributed equally in the whole grayscale range; and finally, the nonlinearity of the constrained histogram is mapped to the main histogram. The experimental averages show that the values of the two metrics, information entropy and MS-SSIM, processed by the algorithms in this paper, are more accurate compared to the other six excellent algorithms.

**Keywords:** histogram segmentation; restricted histogram; histogram equalization; information entropy; narrow image

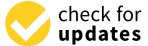



## 1. Introduction

Image enhancement techniques are important fundamental image pre-processing methods. The purposes of image enhancement are to facilitate subsequent image processing and analysis (such as edge detection, modeling, and identification) and to enhance the visual effects of the display device. They have been widely applied in various fields, such as the analysis of satellite images [1,2], video surveillance and analysis [3], face recognition [4], and medical images [5–8].

Among the many image enhancement techniques available, the most widely adopted are based on global HE. Despite its simplicity and effectiveness, this technique has certain inevitable deficiencies, such as excessive brightness, artifacts, and the loss of detailed information [9–12]. Over time, researchers have proposed many improvements to get over these deficiencies. Kim proposed the first improved HE technique in 1997, a brightness-preserving bi-histogram equalization (BBHE) method [13]. In BBHE, the original image histogram is divided into two parts based on the average brightness; then the two parts are equalized. The excessive brightness of images is addressed by enhancing the image through HE. In 1999, a method was proposed for image enhancement based on dualistic sub-image HE (DSIHE) by Wang et al. [14], which segments the original image into two parts with the same-sized areas according to the probability density function of the original image; these two sub-images are equalized. In comparison with the BBHE method, the DSIHE method better preserves the image's brightness and increases the image's information entropy more effectively. In 2003, Chen et al. [15] proposed a recursive mean-separate histogram equalization (RMSHE) algorithm based on BBHE. The sub-images are recursively segmented using the means of the sub-images. The experimental results show that as the mean number of recursive segmentation increases, the mean brightnesses of the output and input images tend to be the same. However, with more number of iterations, the effect of image enhancement declines. In 2007, inspired by the idea of the RMSHE algorithm, a new recursive sub-image histogram equalization (RSIHE) algorithm was proposed by SIM et al. [16]. Although this

algorithm achieves good image compensation, it has the same deficiencies as the RMSHE algorithm—i.e., as the number of recursive calls increases, the effect of the image enhancement degrades seriously. For the purpose of further solving the over-enhancement issues of traditional HE, scholars have proposed a variety of improved HE algorithms, including the adaptive gamma correction with weighting distribution (AGCWD) [17], minimum mean brightness error bi-histogram equalization (MMBEBHE) [18], adaptively increasing the value of the histogram (AIVHE) [19], and bi-histogram equalization using modified histogram bins (BHEMHE) [20].

Besides these aforementioned representative image-enhancement methods that use modified histogram bins based on HE, alternative image-enhancement methods are also proposed based on clipped HE. These alternative methods include the well-known contrast-limited adaptive histogram equalization (CLAHE) algorithm proposed by Karel in 1994 [21]. Inspired by the CLAHE algorithm, Ooi et al. proposed a method based on bi-histogram equalization with a plateau limit (BHEPL) in 2009 [22]. Firstly, in the same way as the aforementioned methods, the input histogram is segmented into two sub-histograms. Subsequently, the two sub-histograms are clipped according to the calculated plateau limit. Not long after the introduction of the BHEPL method, Ooi et al. proposed another method based on the bi-histogram equalization median plateau limit (BHEPLD) algorithm [23], which offers an improvement on the BHEPL method. These two methods effectively avoid over-enhancement. However, both firstly segment the input images into two sub-images and then perform clipping and equalization, respectively. These result in relatively low grayscale stability in some images. In 2014, a novel exposure-based sub-histogram equalization method (ESIHE) was proposed by Singh et al. [24]. The method uses pre-calculated exposure values to split the original image into two sub-images, and crops and equalizes the two sub-histograms. Finally, it merges them to output an enhanced image with information entropy maintained. Santhi et al. used the median to split the input image histogram into four sub-histograms and then clipped the four sub-histograms based on the input image mean while performing histogram equalization separately. This method is called the adaptive contrast enhancement method based on improved histogram equalization (ACMHE), and the algorithm outperforms existing histogram equalization methods in terms of both contrast and structural similarity metrics [25]. Li et al. introduced the normalized coefficient of variation of the histogram to adjust the upper and lower thresholds of the input image histogram, adjusted and clipped the histogram, and finally performed histogram equalization enhancement on the adjusted sub-histograms separately. The algorithm obtained relatively good results for processing the dark areas of the image [26]. Next, Ashiba [27] introduced the idea of homomorphic filtering enhancement into the platform histogram equalization enhancement, and experimental data showed that the algorithm worked well for improving the visualization of night-time infrared images. In 2021, Acharya [28] first smoothed an input histogram using a multinomial curvature fitting function, followed by a resampling process, and finally cropped the histogram using central moment values to achieve suppression of over-enhancement. In the same year, Paul proposed an adaptive histogram equalization algorithm with three platform limits [29]. Although the proposed technique effectively improves artifacts in wide dynamic range images and also increases the image contrast, the artifact improvement and enhancement of low illumination images with narrow dynamic range are less satisfactory. In the following year, Paul et al. proposed a new adaptive enhancement method with dual histogram clips, and the experimental results show that the method can reduce the number of histogram spikes and improve the image contrast, but unfortunately, the method has the same unsatisfactory problems in the processing of low-illumination images with a narrow dynamic range [30].

We present a novel histogram-constrained image-enhancement method in this study that combines the many features of HE-based enhancement technologies mentioned above. In it, the size of the constrained histogram is adjusted by a grey-level probability parameter, thereby controlling the output image contrast and information metrics. This study makes several main contributions as follows.

- We devised a histogram segmentation mechanism for the grey-level probability parameter which splits the input images into two sub-images: the main histogram and the restricted histogram. The grey-scale probability parameter is composed of the number of grey-scale pixels among all pixels in the original histogram, and the effects of this varying parameter on the two metrics of contrast and information entropy of the output histogram are elaborated.
- The main histograms are evenly distributed between *A* and *B*. Adjusting the size of *A* and *B* also changes the output image contrast and average brightness index values. By modifying the main histogram with a uniformly distributed histogram, there will be very few artifacts in the final output image and the brightness of the histogram will become more natural.
- Using the non-linear mapping method given in this paper, the constrained histogram is mapped into the modified master histogram. This aims to reduce the detail loss in the output image, making the main viewing of the enhanced histogram look more detailed and natural.

The organization of the rest of this paper is as follows. The research objective is described and the proposed method is detailed in Section 2. The evaluation metrics are outlined in Section 3. The experimental results and discussions are described in Section 4 using both qualitative and quantitative analyses. Section 5 provides a discussion of the results of the experiment. Finally, the conclusions are drawn in Section 6.

## 2. Research Objective and Method

### 2.1. Research Objective

Previous image-enhancement methods based on HE have deficiencies, including excessive brightness, serious information loss, and the introduction of artifacts into the enhanced image [9–12]. Although many improved methods have been proposed, these methods are only suitable for some images; they cannot achieve the optimum effects when they are used to process narrow-dynamic-range images. Therefore, our purpose was to define a method that improves the contrast of different images (e.g., low-contrast images of relatively narrow dynamic range), reduces over-enhancement, and increases the information entropy of the output histogram to achieve the optimum visual effect of the enhanced histogram.

### 2.2. Proposed Method

Figure 1 shows the flowchart of the grey-scale histogram processing of the proposed technique. The cumulative probability density function, grey-scale probability density function, and histogram of an $M \times N$ grayscale image $I$ are defined as the Equations (1)–(3), respectively.

$$CDF(j) = \sum_{l=0}^{j} q(l) \; j = 0, \cdots, (F-1) \tag{1}$$

$$CDF(j) = \sum_{l=0}^{j} q(l) \; j = 0, \cdots, (F-1) \tag{2}$$

$$H(j) = n_j \; j = 0, \cdots, (F-1) \tag{3}$$

In the above Equations (1)–(3), $j$ is the grey scale value; $F$ is the greyscale image grey level (8-bit greyscale image grey level 256); and $H(j)$, $p(j)$, and $CDF(j)$ represent the number, probability, and cumulative probability of grey scale $j$, respectively.

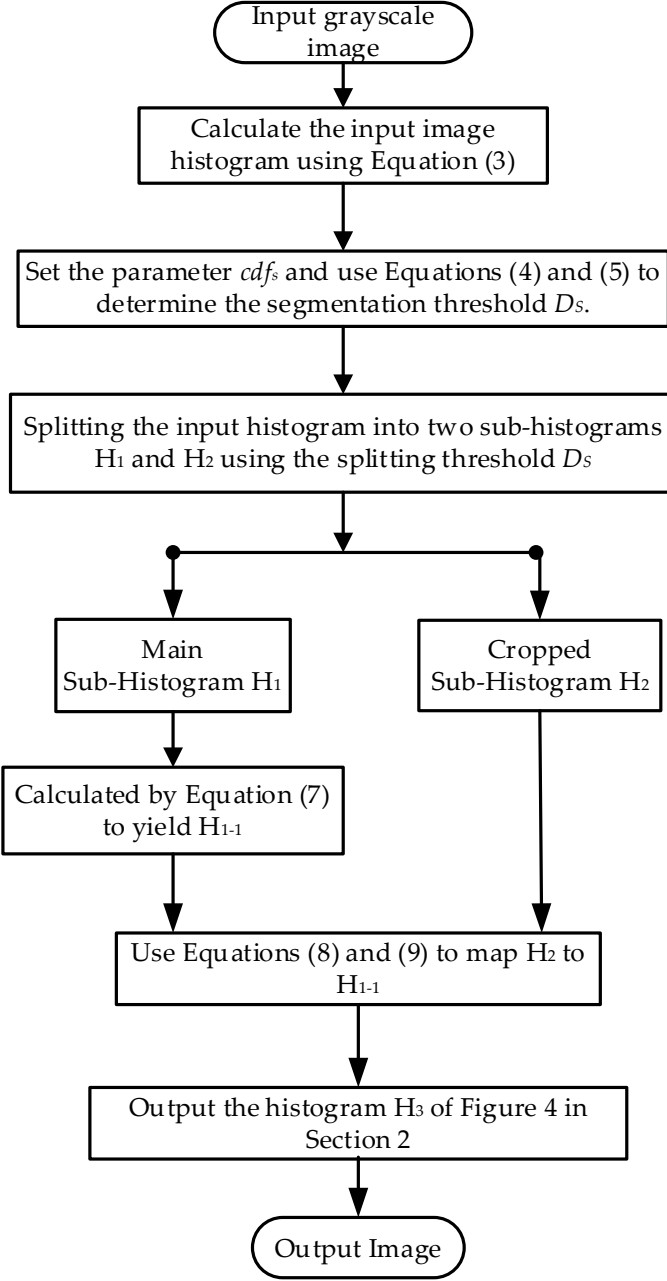

**Figure 1.** Processing flowchart for a grey-scale image.

To elaborate on the proposed algorithm, we assume that the input image has a grey level of 21 and a range of pixel grey level variations [0 20]. Table 1 shows the pixel grey-level values and the corresponding quantitative relationships, which are distributed schematically in Figure 2a.

**Table 1.** Relationship between gray level and number of pixels.

| Variable | Histogram Values | | | | | | | | | |
|---|---|---|---|---|---|---|---|---|---|---|
| Gray level | 1 | 3 | 4 | 6 | 7 | 8 | 12 | 16 | 17 | 20 |
| Number of pixels | 3 | 4 | 7 | 14 | 7 | 4 | 10 | 12 | 3 | 16 |

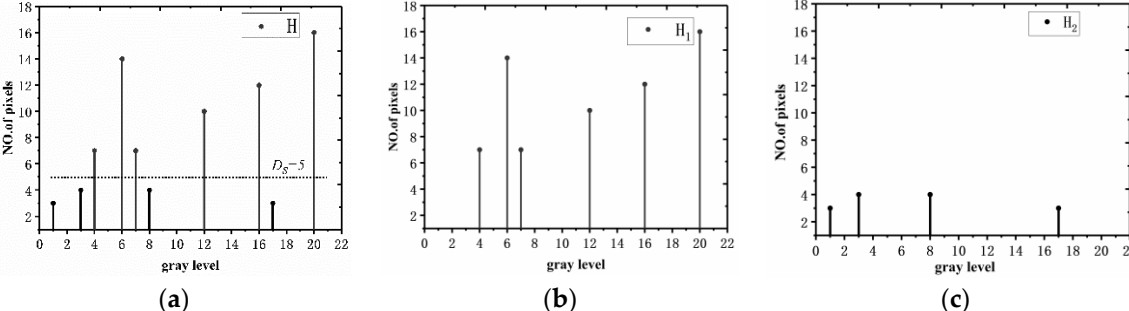

**Figure 2.** Histogram segmentation diagram: (**a**) input image histogram, (**b**) sub-histogram $H_1$, (**c**) sub-histogram $H_2$.

The histogram-limited contrast enhancement method involves three parts: segmenting the input image histogram into $H_1$ and $H_2$; using $H_1$ to establish the uniformly distributed sub-histogram, $H_{1-1}$; and mapping the restricted sub-histogram $H_2$ into the sub-histogram $H_{1-1}$ to create a histogram for the output image. The three main parts of the algorithm are described in detail with the assumption that the segmentation threshold $H_T = 5$, and 10 discrete histograms of effective gray levels are set to the same time in the range of [0 20]. These parts are described in three sections below in detail.

- The first part is the segmentation of the original histogram. Assuming $cdf_s$ is the set to a cumulative probability partition value, the corresponding histogram number partition value $D_s$ is calculated cyclically according to Equation (4). For the actual calculation, the initial value $D_S = 0$ is set first, and whether $cdf(j) \geq cdf_S$ is satisfied is judged after $cdf(j)$ is calculated by Equation (5). If the condition is satisfied, then $D_S = D_S + 1$; otherwise, the value of $D_S$ remains unchanged.

$$D_S = \begin{cases} D_S + 1 & cdf(j) > cdf_s \\ D_S & \text{Otherwise} \end{cases} \qquad (4)$$

$$cdf(j) = \frac{\sum\limits_{j=0}^{F-1} (H(j) > D_s)}{\sum\limits_{j=0}^{F-1} H(j)} \qquad (5)$$

where $cdf_S$ is the cumulative probability splitting parameter, whose value is in the range of [0.9 1]; $D_S$ is the histogram number splitting threshold corresponding to the $cdf_S$ parameter; $cdf(j)$ is the cumulative probability density of $H(j) > D_S$.

In terms of $D_S = 5$, the original histogram is split into two sub-histograms, $H_1$ and $H_2$, which are called the main histogram and the limited histogram, respectively. The segmentation diagram is shown in Figure 2, and the expression for the histogram segmentation is Equation (6).

$$H = \begin{cases} H_1 & H > D_S \\ H_2 & H \leq D_S \end{cases} \qquad (6)$$

- The second part is the uniform distribution of sub-histogram $H_1$. Suppose [$A$ $B$] is the histogram grayscale range of the input image, in our schematic $A = 0$, $B = 20$; then, the sub-histogram $H_1$ is uniformly distributed in this range; the uniformly distributed histogram is recorded as $H_{1-1}$. The uniform distribution diagram is shown in Figure 3, and the uniform distribution equation of the histogram is defined as the Equation (7).

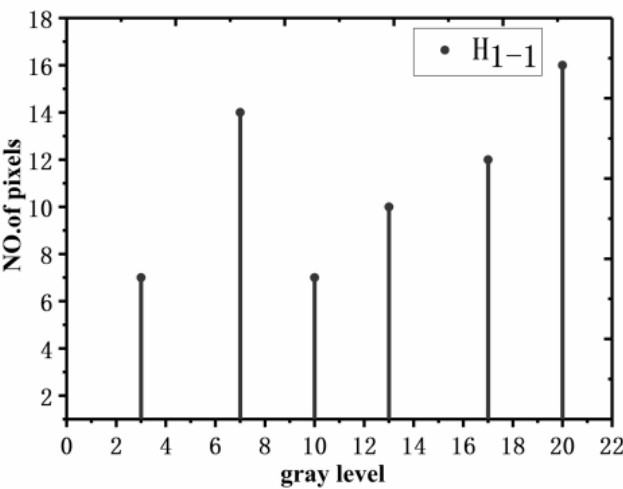

**Figure 3.** The uniform distribution of histogram $H_{1\text{-}1}$.

$$T_F(j) = round\left(\frac{B-A}{j_{\max}}\right) \times j + A \text{ for } j = 1, \cdots, j_{\max} \tag{7}$$

where $T_F$ is a uniformly distributed transformation function for the grey values of the main histogram $H_1$. $j_{\max}$ is the number of sub-histograms $H_1$, and $j_{\max} \neq 0$. $j$ is a number variable. $A$ and $B$ are the limits of the gray value range $[A, B]$ ($A = 0$ and $B = 255$ are satisfactory in the 8-bit gray image), and round () is an integer function.

- The third part is to map the restricted sub-histogram $H_2$ into the sub-histogram $H_{1\text{-}1}$. Assume $i$ is the index gray value variable of the main histogram $H_1$ and $I$ is the index gray value of restricted histogram $H_2$. The non-linear mapping process consists of 3 main steps. First, to find the grey value $D_I$ with the smallest difference in value and its corresponding index value $t$, we select the grey value $I$ from the sub-histogram $H_2$ and compare it with all the grey values in the $H_1$ histogram, in turn, calculated as shown in Equation (8). Taking the input image histogram $H$ as an example, if we choose $I = 3$, we find that only the grey value 4 in $H_1$ is closest to 3. At this point, we can determine $D_I = 4$, $t = 1$. Secondly, we calculate the new grey value $I_t$ in $H_{1\text{-}1}$ after uniform distribution of grey values according to Equation (9). After the calculation, we find that $D_I = 4$ in $H_2$ has become the new grey value 3 in $H_{1\text{-}1}$. Finally, all $D_I$ grey values of the restricted sub-histogram $H_1$ are mapped to the sub-histogram $H_{1\text{-}1}$. In this example, $D_I = 4$ is mapped to $I_t = 3$, and Figure 4 represents the final mapping process and results.

$$D_f(I) = \underset{i,I \in \{0, \cdots, F-1\}}{\arg\min} |i - I| \tag{8}$$

$$I_t = round(\frac{B-A}{j_{\max}}) \times t + A t = 1, \cdots, j_{\max} \tag{9}$$

In Equations (8) and (9), $I$ and $i$ represent the grey values in $H_1$ and $H_2$, respectively; $D_I$ and $t$ are the grey values and index values closest to $I$ in $H_2$, and $I_t$ is the new grey scale value after $D_I$ mapping to $H_{1\text{-}1}$.

After the third part of the non-linear mapping process, the artefacts in the output image can be greatly reduced, making the output image look subjectively more visually satisfying and richer in detail.

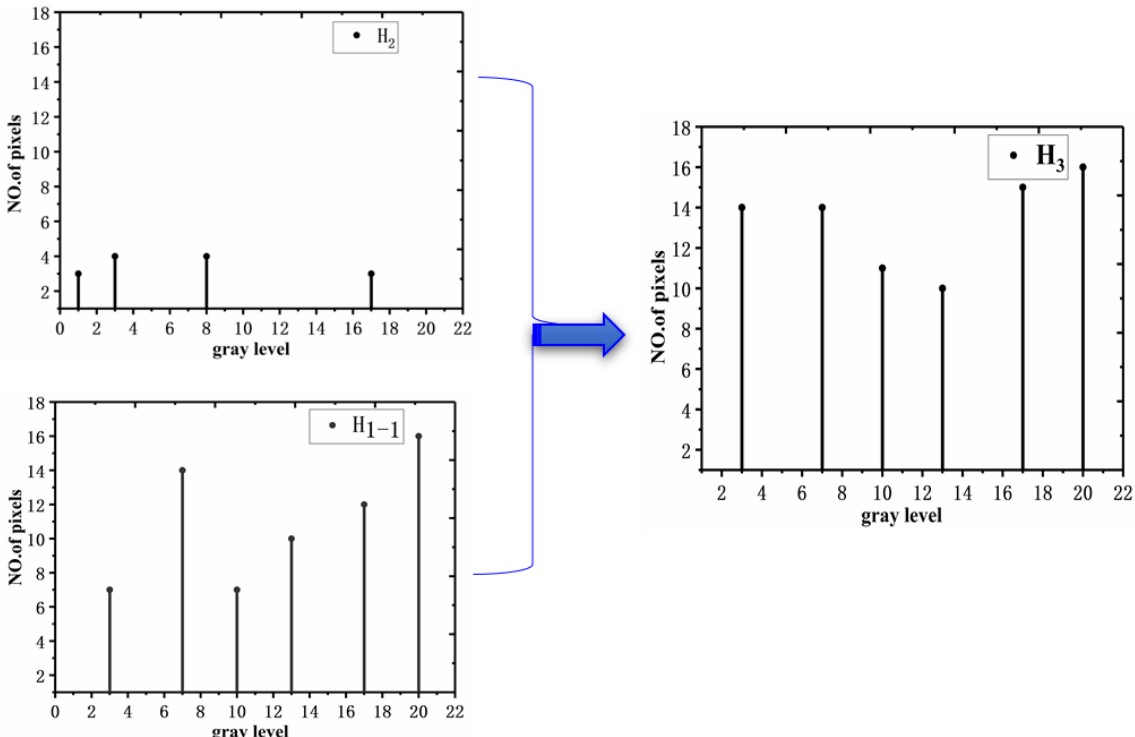

**Figure 4.** Mapping process of the restricted histogram.

### 3. Evaluation Metrics

(i) Information entropy [31]

In an image, information entropy is a measure of the level of detail. If the enhanced image has a higher level of detail, the image needs a higher information entropy value. The entropy *E* can be calculated by the grey-scale probability density *q(j)* using Equation (10).

$$E = -\sum_{j=0}^{F-1} q(j) \log_2 q(j) \tag{10}$$

(ii) Contrast [32,33]

Contrast is an important indicator of the level of contrast in an image. A higher contrast level indicates a better image enhancement effect. Contrast is calculated using the Equation (11).

$$C = 10 \log_{10}\left[ \left(\frac{\sum I_{en}(u,v)}{M \times N}\right)^2 - \frac{\sum I^2_{en}(u,v)}{M \times N} \right] \tag{11}$$

In the above equation, *C* is the contrast variable, $I_{en}$ is the output-enhanced image, and *u* and *v* are the numerical variables of the horizontal and vertical coordinates of the $I_{en}$ image, respectively.

(iii) Peak signal-to-noise ratio [26,34]

PSNR is a technical index that measures the noise rejection performance in image processing algorithms. A higher PSNR indicates more effective noise suppression. PSNR is calculated using Equation (12).

$$PSNR = 10\lg\left[ \frac{M \times N \times (2^8 - 1)^2}{\sum_{\alpha}\sum_{\beta}|I(\alpha,\beta) - I_{en}(\alpha,\beta)|^2} \right] \tag{12}$$

where $I_{en}$ $(\alpha,\beta)$ represents the enhanced image, $M$ is the number of rows, and $N$ is the number of columns in the input image $I(\alpha,\beta)$.

(iv)  MS-SSIM

MS-SSIM (multiscale structural similarity) is a multi-scale method for evaluating image quality and is an extension of the SSIM metric by Wang et al. [35]. MS-SSIM performs contrast comparison and structural comparison at each scale *i*, and a luminance comparison is calculated only at the final scale *M*. The MS-SSIM calculation formula is defined as Equation (13).

$$MS - SSIM(X,Y) = [l_M(X,Y)]^{\alpha_M} \prod_{i=1}^{M}[C_i(X,Y)]^{\beta_i}[S_i(X,Y)]^{\gamma_i} \tag{13}$$

where *i* is the scale variable; *M* is the final scale; *X* and *Y* are the image matrices; and $\alpha$, $\beta$, $\gamma$ are the corresponding weight indices of the respective comparison functions. $C_i(X,Y)$, $S_i(X,Y)$, and $l_M(X,Y)$ are the contrast comparison function, the structure comparison function, and the luminance comparison function, respectively. *MS-SSIM(X,Y)* is defined as a multi-scale similarity function.

## 4. Experiment Results and Analysis

The experimental images in this paper are mainly from well-known image datasets available on the Internet, such as USC-SIPI (https://sipi.usc.edu/database/database.php?volume=sequences, accessed on 15 October 2022) and BSD300 (https://www2.eecs.berkeley.edu/Research/Projects/CS/vision/bsds/, accessed on 15 October 2022), and to a lesser extent from images taken and produced by ourselves. These online images are not used for any commercial purpose; they are used to test algorithms. The running environment for the experiments was a Windows 10 operating system on an Intel Pentium CPU G860 with 3.0 GHZ and 16.0 GB RAM. All algorithms in the paper were written in code and implemented using MATLAB2020b.

### 4.1. Impacts of the Parameters A and B on Brightness and Contrast

The grayscale range [*A*, *B*] determines the brightness of the image enhanced with the novel algorithm. In other words, the brightness levels of the low and high-gray areas of the enhanced image are determined by the values of *A* and *B*, respectively. Figure 5a shows the input image "Tractor". Figure 5b–d were obtained after the enhancement of Figure 5a using the proposed algorithm when the grayscale ranges [*A*, *B*] were set to [0, 255], [50, 255], and [0, 150], respectively, and the $cdf_s$ was set to 0.9999.

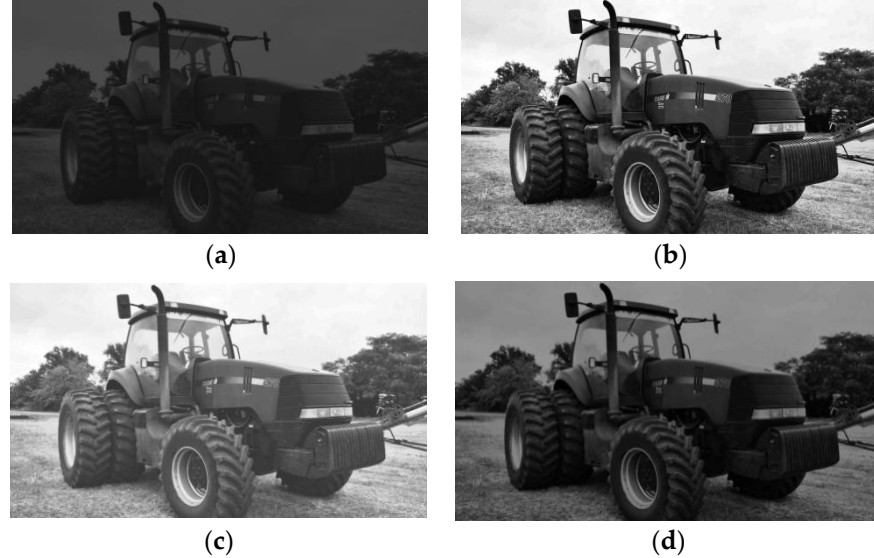

(a)                 (b)

(c)                 (d)

**Figure 5.** Impacts of *A* and *B* on brightness and contrast: (**a**) original; (**b**) [0, 255]; (**c**) [50, 255]; (**d**) [0, 150].

The difference between *A* and *B* controls the brightness of the image and affects the other metrics of the output image. The graph below shows the variation curves of the $B-A$ differences in the output image's indexes. One-hundered images were tested under the following assumptions: when $cdf_s$ = 0.9999, initially, *A* = 0 and *B* = 255. The lower limit *A* of the grayscale range increased from 0 at an increment of 10, and the upper limit *B* of the grayscale range decreased from 255 at a decrement of 10, while complying with the condition of $B-A$ > 0.

Figure 6 shows that as the $B-A$ difference increases, the contrast and mean brightness of the enhanced image also increase, which is consistent with the visual perception of the human eye in Figure 5.

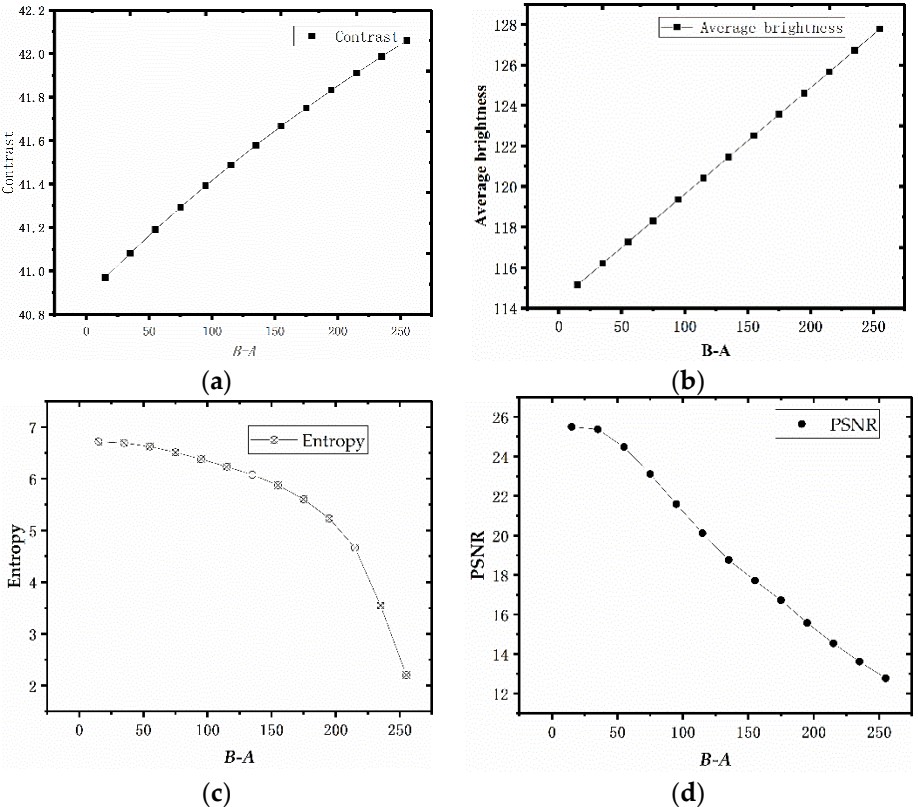

**Figure 6.** The plot of output image metrics versus $B-A$: (**a**) contrast; (**b**) mean brightness; (**c**) entropy; (**d**) PSNR.

### 4.2. Impacts of cdf_s Parameters on Algorithm Performance Metrics

To prevent the distortion of the enhanced image, the parameter $cdf_s$ is usually valued within the range of [0.9, 1]. The smaller the $cdf_s$ value is, the larger the contrast of the output image is, and vice versa. The mean brightness of the enhanced image decreases as the $cdf_s$ value increases. In contrast to the changes in the trend curves of the first two metrics, the values of the performance metrics of PSNR and information entropy gradually increase as the values of the $cdf_s$ parameters increase. One-hundred images were tested, and the output image contrast and average brightness were calculated. Those two indicators varied with the parameter $cdf_s$, as shown in Figure 7. The grayscale range of the tested images was [0, 255], and the parameter $cdf_s$ increased gradually within the range of [0.9, 1].

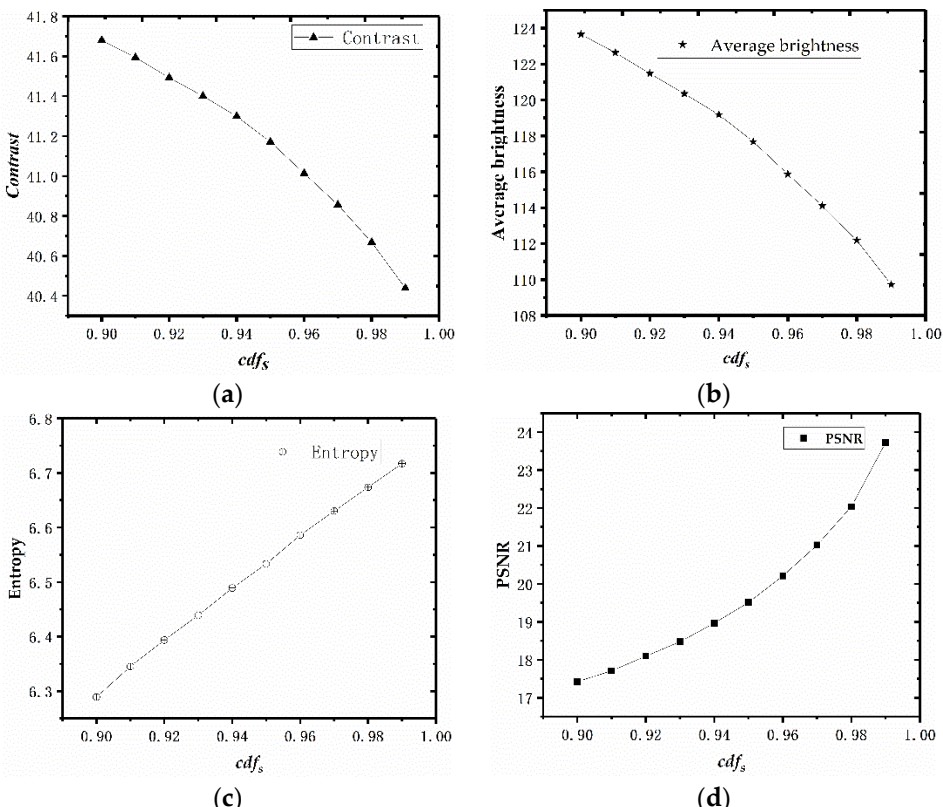

**Figure 7.** Curves of the effects of cdfs parameters on image performance metrics: (**a**) contrast; (**b**) mean brightness; (**c**) entropy; (**d**) PSNR.

### 4.3. Comparison of Algorithms

The proposed algorithm can effectively improve the image information entropy and contrast for degraded images (such as narrow-dynamic-range images). To verify the algorithm's performance, the proposed algorithm was experimentally tested on most of the narrow dynamic images in this paper. Due to the length of the paper, it is not possible to list all the images in this paper. Only three different representative narrow dynamic images were randomly selected from the public image database for description. These images include the narrow-dynamic-range image "Tractor" with low grayscale, the narrow-dynamic-range image "fish" with medium grayscale, and the narrow-dynamic-range image "bridge" with high grayscale. Six excellent algorithms, including Ref. [30], BHEMHE, RMSHE, DSIHE, AGCWD, and BHEPL, are compared with the proposed algorithm. The parameters of the proposed algorithm were set as follows: $cdf_s = 0.9999$, $A = 0$, and $B = 255$. The three original images and their corresponding grayscale distribution histograms are shown in Figure 8.

#### 4.3.1. Narrow-Dynamic-Range Image "Tractor" with Low Grayscale

The performances of the seven algorithms in processing the narrow-dynamic-range images with low grayscale were tested. The images enhanced with these algorithms are shown in Figure 9, and the experiment results are summarized in Table 2.

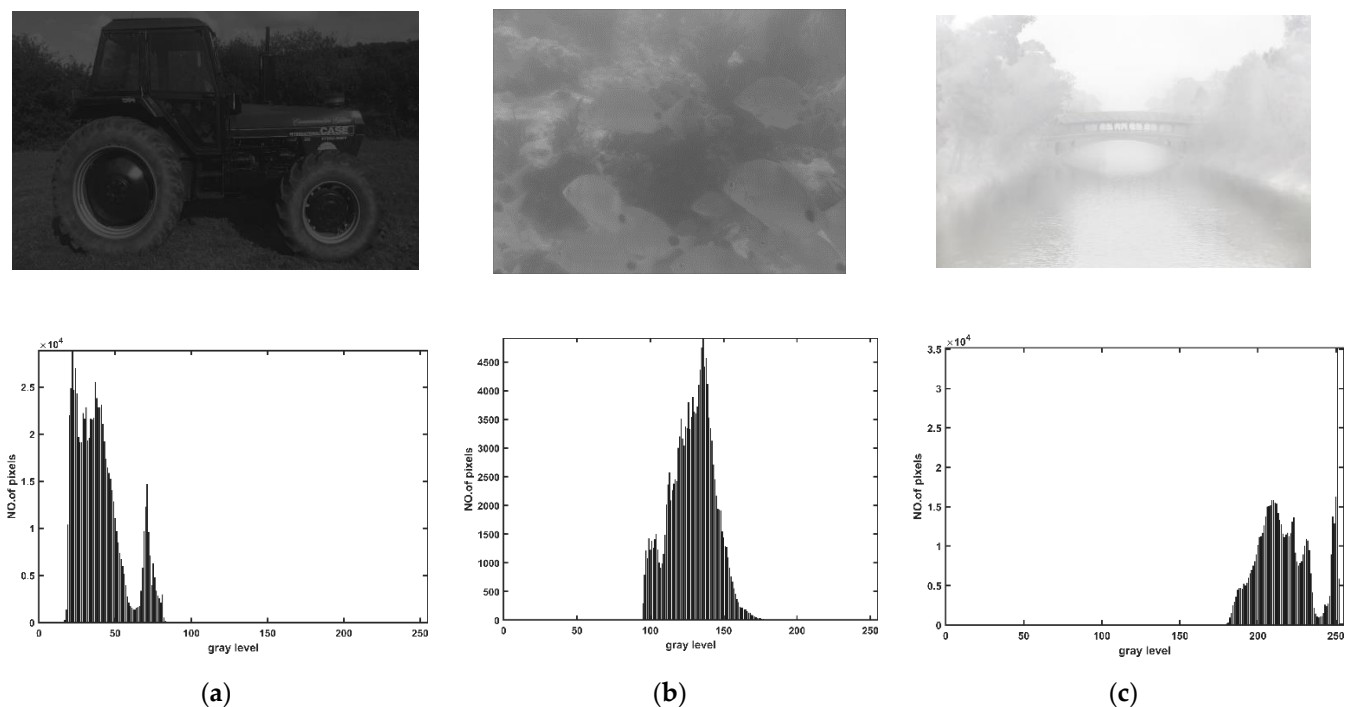

**Figure 8.** Original images and their corresponding histograms: (**a**) Image "Tractor"; (**b**) Image "Fish" (**c**) Image "Bridge"

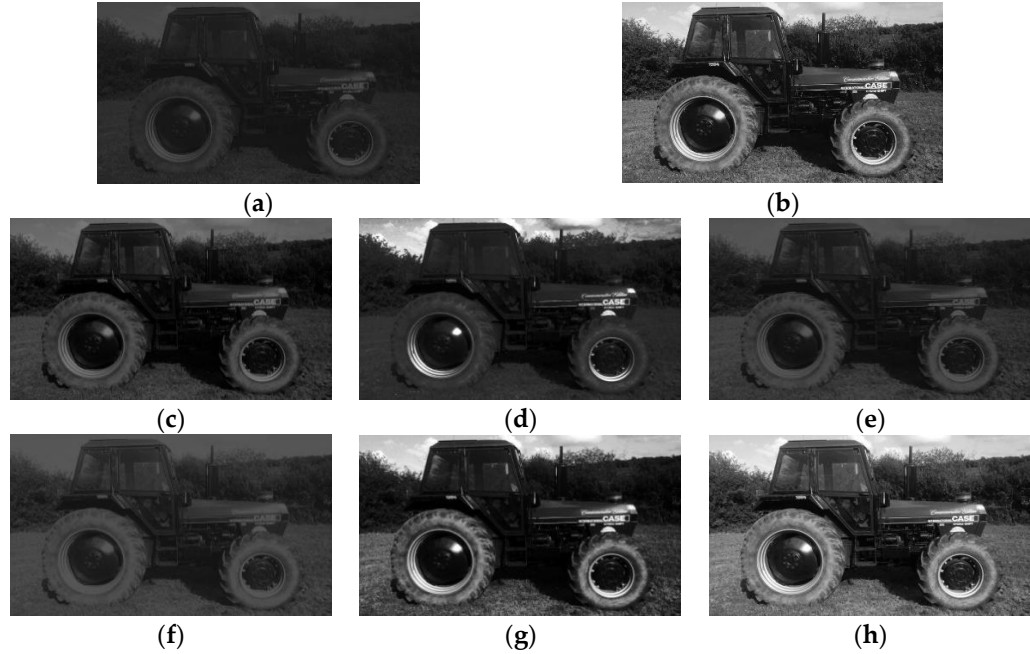

**Figure 9.** Image named "Tractor" enhanced with several algorithms: (**a**) original image, "Tractor"; (**b**) Ref. [30]; (**c**) BHEMHE; (**d**) RMSHE; (**e**) DSIHE; (**f**) AGCWD; (**g**) BHEPL; (**h**) proposed method.

**Table 2.** Experimental data for the low-grayscale image "Tractor".

| Techniques | Entropy (bit) | PSNR (dB) | C (dB) | MS-SSIM |
|---|---|---|---|---|
| Input image | 5.6278 | — | 31.19 | 1 |
| BHEMHE | 5.6254 | 23.97 | 32.07 | 0.8768 |
| RMSHE | 5.5856 | 15.92 | 33.81 | 0.8364 |
| DSIHE | 5.4967 | 28.21 | 32.43 | 0.9389 |
| AGCWD | 5.3453 | 23.58 | 34.34 | 0.9377 |
| BHEPL | 5.5817 | 13.44 | 35.84 | 0.6624 |
| Ref. [30] | 5.6268 | 12.49 | 38.00 | 0.6477 |
| Proposed | 5.6272 | 11.49 | 39.19 | 0.6546 |

For the narrow-dynamic-range image "Tractor", the contrast and brightness of the original image are relatively low, resulting in limited visibility of image details and poor visual effects. Figure 9 shows that all seven algorithms can enhance the original image. The algorithm proposed in this paper had the most significant enhancement effect, followed by that of the Ref. [30] algorithm. This conclusion is supported by the contrast indicator data in Table 2. The contrast of the images enhanced with the other algorithms is lower, resulting in unclear details in the dark areas. In comparison, the experimental data in Table 2 demonstrate that the information entropy and contrast values of the images enhanced by the algorithm in this paper are the largest. This indicates the proposed algorithm is superior at highlighting the details in dark areas and increasing the contrast. However, the PSNR and MS-SSIM achieved by the proposed algorithm during this experiment were the lowest, indicating that this algorithm does not perform well concerning noise suppression.

### 4.3.2. Narrow-Dynamic-Range Image "fish" with Medium Grayscale

This section provides an experimental test of narrow-dynamic-range images that were in the middle of the histogram.

For the visual effects of the enhanced images in Figure 10, the differences between the brightness and the darkness of the images processed by DSIHE, Ref. [30], and AGCWD are small, and the details are not clear. In the image "fish" enhanced with RMSHE, the loss of detailed information in the fish's bodies is serious, and the image is also distorted to a certain extent in terms of the visual effect. Although the image enhanced with the BHEMHE algorithm has strong contrast, this algorithm also causes over-enhancement; this resulted in the loss of details in the excessively bright areas. The images enhanced with the BHEPL algorithm and the algorithm proposed in this paper are similar concerning contrast and brightness; they provide better visual effects in comparison to the other five algorithms. The experiment results listed in Table 3 show that there was no great difference in the contrast of the images enhanced with the seven algorithms, which is approximately 43, indicating that the objective image enhancement effects of these seven algorithms are consistent. As shown in Table 3, the information entropy metric shows that the image processed by this algorithm has the highest value of information entropy (5.8821), indicating that the image processed by this algorithm is the richest in detail. The PSNR achieved by the proposed algorithm ranks fifth, indicating a relatively low level of noise suppression.

**Table 3.** Experimental data for the medium grayscale image "fish".

| Techniques | Entropy (bit) | PSNR (dB) | C (dB) | MS-SSIM |
|---|---|---|---|---|
| Input image | 5.8827 | — | 42.13 | 1 |
| BHEMHE | 5.5462 | 11.79 | 42.29 | 0.4937 |
| RMSHE | 5.8231 | 16.86 | 42.14 | 0.6749 |
| DSIHE | 5.7467 | 24.62 | 42.66 | 0.8972 |
| AGCWD | 5.7231 | 20.30 | 43.56 | 0.9397 |
| BHEPL | 5.7756 | 14.71 | 42.69 | 0.5952 |
| Ref. [30] | 5.8789 | 16.55 | 43.47 | 0.7744 |
| Proposed | 5.8821 | 15.82 | 42.37 | 0.7309 |

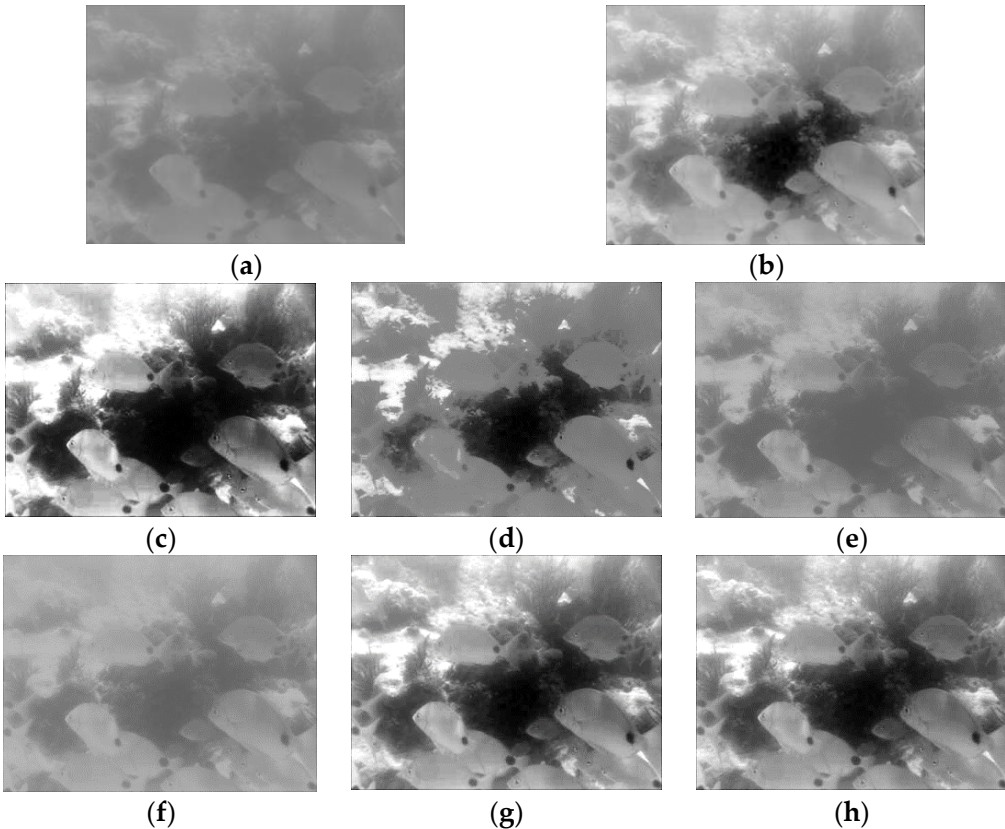

**Figure 10.** Images of "fish" enhanced with several algorithms: (**a**) original image, "fish"; (**b**) Ref. [30]; (**c**) BHEMHE; (**d**) RMSHE; (**e**) DSIHE; (**f**) AGCWD; (**g**) BHEPL; (**h**) proposed method.

### 4.3.3. Narrow-Dynamic-Range Image "Bridge" with High Grayscale

All algorithms enhance narrow-dynamic-range images in the high-grey-level band, and the subjective results and quantitative analysis are presented below.

Table 4 shows that the image processed with the proposed algorithm has the lowest contrast (42.66) but has the optimum subjective visual appearance for human eyes. Figure 11 shows that the proposed algorithm enhances the original image without any signs of over-enhancement. In comparison to the images enhanced with the other six algorithms, the image processed with the proposed method is smoother, and has clearer details in the bright and dark areas, including the ripples of river water, trees on both banks of the river, and tree leaves. In contrast, the images enhanced with Ref. [30], BHEMHE, RMSHE, DSIHE, AGCWD, and BHEPL algorithms all show signs of over-enhancement. For example, the ripples of the river water in the bright areas were lost during the image enhancement. The image processed with the RMSHE algorithm is distorted to a certain extent and has a poor appearance. For example, the trees on the riverbanks appear as if veiled by white satin. Based on the objective data presented in Table 4, the information entropy value (5.8725) of the image processed with the method of this paper is clearly the highest. This indicates the image processed with the proposed algorithm contains the most abundant level of detail. Unfortunately, the PSNR (8.46) and contrast ratio (42.66) of the image processed by the proposed algorithm have relatively low values. This indicates the proposed algorithm is inferior to the other six algorithms in terms of noise suppression and contrast enhancement.

**Table 4.** Experimental data for the high-grayscale image "bridge".

| Techniques | Entropy (bit) | PSNR (dB) | C (dB) | MS-SSIM |
|---|---|---|---|---|
| Input image | 5.8731 | — | 46.76 | 1 |
| BHEMHE | 3.8784 | 10.03 | 45.06 | 0.6085 |
| RMSHE | 5.7085 | 13.58 | 45.82 | 0.7787 |
| DSIHE | 5.6787 | 32.11 | 46.69 | 0.9756 |
| AGCWD | 5.4201 | 24.89 | 47.27 | 0.9785 |
| BHEPL | 5.5614 | 12.81 | 45.36 | 0.7131 |
| Ref. [30] | 5.8470 | 20.95 | 46.60 | 0.9214 |
| Proposed | 5.8725 | 8.46 | 42.66 | 0.7899 |

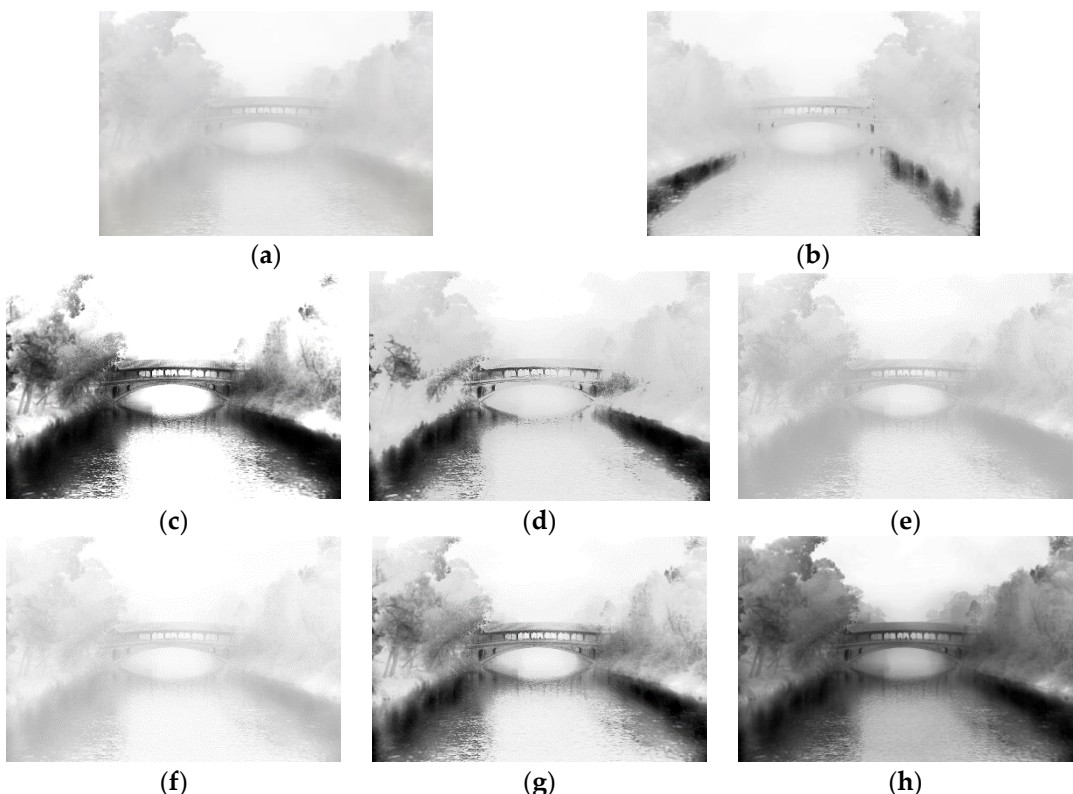

**Figure 11.** Results of several algorithms for "bridge" image enhancement. (**a**) original image, "bridge"; (**b**) Ref. [30]; (**c**) BHEMHE; (**d**) RMSHE; (**e**) DSIHE; (**f**) AGCWD; (**g**) BHEPL; (**h**) proposed algorithm.

### 4.3.4. Average Performance Metric for All Algorithms

To further verify the excellent performance of the proposed method in image enhancement in terms of the evaluation metrics, the performances of different algorithms in processing the 300 images were tested. We chose the well-known image dataset BSD300 for the average data calculation, with image resolutions of $481 \times 321$ and $321 \times 481$. Table 5 shows the average values of the performance indices of different algorithms for the 300 images. The best value for each analysis is in boldface.

**Table 5.** Average values of quantitative analyses for 300 test images.

| Techniques | Entropy (bit) | PSNR (dB) | C (dB) | MS-SSIM |
|---|---|---|---|---|
| Input images | 7.1273 | — | 40.35 | 1 |
| BHEMHE | 6.4441 | 18.13 | 40.51 | 0.8358 |
| RMSHE | 6.9712 | 24.66 | 40.56 | 0.9272 |
| DSIHE | 6.9189 | 19.61 | 41.39 | 0.8653 |
| AGCWD | 6.8224 | 14.61 | 43.45 | 0.8872 |
| BHEPL | 5.9408 | 16.80 | 41.12 | 0.8164 |
| Ref. [30] | 7.1151 | 31.27 | 40.68 | 0.9873 |
| Proposed | 7.1266 | 30.90 | 40.74 | 0.9905 |

The objective evaluation data based on the processing of 300 images show that the information entropy of the output image processed with the proposed method has the largest value (7.1266), which further verifies that among the seven algorithms studied in this paper, this algorithm has the best performance in highlighting image details. Similarly, the MS-SSIM metric, with a numerical size of 0.9905, ranks first and is also significantly better than the other algorithms. In terms of noise suppression, the Ref. [30] algorithm achieved the highest PSNR (31.27), and the PSNR for the proposed algorithm is 30.90, ranking second among all of the algorithms. This indicates the proposed algorithm also has strong performance in terms of noise suppression. Although the average contrast ratio (40.74) enhanced by this algorithm is relatively small, the images enhanced with this algorithm are free from any excessive brightness, over-enhancement, and artifacts, making the enhanced images clearer and more natural.

## 5. Experimental Discussions

In our experiments, we described the effect of the grey-scale range [*A B*] on the output image's metrics, which in general take a value within [0 255]. In addition, we gave trend curves for the main metrics, such as information entropy and PSNR, with the main control parameter $cdf_s$. How the $cdf_s$ takes values in [0.9 1] depends on what the user wants to highlight. If you want to have a relatively high contrast index, then $cdf_s$ can take a smaller value. It should be noted that it is best not to go below 0.9; otherwise, the output image will appear distorted. In most cases, we take a compromising approach, which means that the output image is not only of good contrast but also rich in detail.

We have also enhanced defective images with different grey levels (e.g., the low-light image "Tractor", the underwater image "fish" and the haze image "bridge"). The main purpose of the experiment was to see if the proposed algorithm could render dark details clearly and if there were any problems with over-enhancement (over-brightening). However, in terms of defect image processing, the algorithm in this paper is somewhat lacking in both *PSNR* and *MS-SSIM* outcomes.

Section 4.3.4 gave the average data of 300 images processed by all the algorithms. The experimental data show that the algorithm proposed in this paper obtained the first ranking in terms of multi-scale structural similarity (MS-SSIM) index and information entropy compared to the other six excellent algorithms.

## 6. Conclusions

In this paper, an effective, histogram-limited contrast enhancement method was proposed. This method can adaptively adjust the narrow-dynamic-range images with low, medium, or high gray levels to the maximum dynamic range [0, 255]. It can adaptively increase the image's contrast, reduce over-enhancement, and increase the information entropy of the output image to achieve the optimum visual effect for human eyes. The proposed algorithm is innovative in that it controls the size of the limited histogram to be segmented using the cumulative probability density threshold PT of the original histogram and maps the grayscale of the limited histogram to the grayscale of the uniformly distributed histogram according to the mapping rule. The experimental results show that

when PT = 0.99, the proposed method can obtain an enhanced image with the largest value of information entropy. The contrast and the level of detail in the output image may be adjusted by selecting different values of the parameter PT of the proposed algorithm, to meet the users' needs. Tables 2–5 show that the proposed method has a relatively low value of noise rejection performance for the output image when processing the image. This indicates that the algorithm proposed in this paper needs further research and improvement for enhancing the noise processing of images.

**Author Contributions:** Data curation, X.F. and J.W.; Methodology, X.F. and J.W.; Formal analysis, H.W; Software, H.W.; Funding acquisition, C.X.; Project administration, C.X.; All authors have read and agreed to the published version of the manuscript.

**Funding:** This research was funded by the National Key Research and Development Program of China, grant number 2016YFD0700400.

**Data Availability Statement:** Not applicable.

**Conflicts of Interest:** The authors declare no conflict of interest.

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
