# Peer review of "Contrast-Controllable Image Enhancement Based on Limited Histogram"

_electronics, doi:10.3390/electronics11223822_

Round 1
Reviewer 1 Report
The authors have proposes a histogram-constrained and contrast-tunable Histogram Equalization technique for digital image enhancement. After reviewing the manuscript, it is observed that there are minor concerns to be addressed:
1. The authors need to cite the five algorithms, including BHEMHE, RMSHE, DSIHE, AGCWD and BHEPL which are compared with the proposed algorithm in the tables.
2. The authors need to thoroughly check the manuscript for the grammar issues for instance in lines 232-237, it is not in the proper format. The authors need to fix these type of things in the whole manuscript.
3. The authors need to add the discussion section in which they can address the limitations of their proposed method along with the advantages, for the reader's interest.
Reviewer 2 Report
The authors need to incorporate the latest performance measuring metrics as published in reference [27] which includes feature similarity index measure (FSIM), spectral residual similarity index measure (SR-SIM), gradient magnitude similarity deviation (GMSD), visual saliency-induced index (VSI) and multi scale structural similarity index measure (MS-SSIM). Then only comparison can be made with the state-of-the-art method(s).
The source of the dataset is also not clear. Please cite the link(s) of the dataset. Have you been given permission by the owner(s) to use them in your paper?
At the moment only Entropy implies superiority of the method but it was not even compared with the state-of-the-art (e.g., Ref. [27]).
The number of images tested is also low compared to reference [27] which tested on 300 images.
Reviewer 3 Report
The article presents an interesting technique for image processing. I have the following suggestions for improving this article further.
1. The variables in eq. 1-3 should be explained before they are used in the equations. Same for forthcoming equations as well. What is Q(j), q(j), DS, D_s, M, N, Q_s, and so on?
2. Various blocks of the flowchart should be linked with the respective equations and sections to help better understanding. The flowchart should contain more details.
3. Why can't we use a graph with various curves/trends shown differently by respective legends in Fig. 2? Moreover, these images are blurred and the background color should preferably be removed.
4. Captions for Tables 1, 2, and 3 are similar but they correspond to different results. This is confusing as well as misleading. Please use other captions to elaborate on the type of data shown in these tables.
5. It is noted that the proposed algorithm shows improved results for Entropy(bit) and C(dB) for tractor images whereas the results for fish and bridge are only better in terms of Entropy. Please elaborate on this. Similarly, for data in Table 4, only Entropy is better for the proposed method. If the proposed method is unable to produce better performance in terms of the considered indices, what is the usefulness of presenting this technique?
6. I think the introduction section should contain more information in order to relate the proposed work to the state-of-the-art. This section needs more attention.
Thanks
Reviewer 4 Report
The manuscript by Fan et al. investigates the enhancement of low contrast images using an histogram-constrained algorithm in which the initial image is split into two sub-images whose histograms are redistributed before recombination. The authors show with various examples that the resulting images possess a higher information entropy than other HE algorithms, albeit at the price of a lower peak signal to noise ratio.
Simple and efficient HE techniques are important for image processing and the present work attempts at proposing an alternative to existing algorithms, which can be inefficient with low contrast images. However, the present work appears to me more like a trial and error numerical study than a thorough and rigorous demonstration of the efficiency of a new method. My concerns have to do with both the presentation, which is lacking in many places, and with the logical way the study is conducted. Substantial work would be required for this work to be publishable.
Below is a list of points, both minor and major, which should be addressed.
-Many quantities (e.g. "q_j", "n_j" in eqs. 1-3) are not introduced.
-The logic behind introducing the redistributioin of H1 as per eq. 7 is not explained. Neither is the nonlinear mapping of eq. 8.
-Both in the abstract and the text (l. 204) the authors refer to "subjective" enhancement. This does not belong to a scientific discussion.
-In sec. 4 why do the authors examine the effects on their A and B parameters only on the contrast and mean brightness, but not on the other metrics? The resulting variations are by the way trivial (figs. 6 and 7) and could be simply mentioned in the text.
-When comparing the metrics of the resulting images the authors should include those of the initial images, so that one can quantitatively judge the enhancement.
-In table 4 the authors report some statistics over 80 images, but the distributions of the metrics of the initial images should be discussed, otherwise one cannot judge of the general applicability of the algorithm.
-Generally (and this goes together with the logic behind the proposed algorithm), the authors should attempt at interpreting their results, in particular the systematic improvement in information entropy but poor PSNR. I imagine that this could be illustrated on simple examples with simple (binary?) histograms, for which the used transformations and subimages are trivial.
Round 2
Reviewer 2 Report
The authors have addressed the major issues that I have highlighted. While the proposed method does not outperform the other state-of-the-art techniques in all metrics, it does provide better results in terms of entropy and MS-SSIM.
Author Response
Dear Reviewer,
I have carefully read and analyzed the comments you have given me. These days, I have tried my best to do a good job of revising and digging deeper. Now I have finished the revision work and received your approval. A "revised" draft with some minor changes has been submitted as requested by the editor. Thank you again for the guidance you have given us on the manuscript.
Kind regards,
Xin Fan
Reviewer 3 Report
Paper seems OK
Author Response

(The authors gave the same response as above.)

Reviewer 4 Report
The authors have to some extent taken my comments into account and improved the presentation of their manuscript, which is close to being in a publishable form. The authors should still ensure that all quantities introduced are properly defined. This is, for instance, not the case for all the functions appearing in eq. 13, introducing the new MS-SSIM metric. The authors should at least refer to the relevant literature.
